# Construction of a Modified Clip Cage and Its Effects on the Life-History Parameters of *Sitobion avenae* (Fabricius) and Defense Responses of *Triticum aestivum*

**DOI:** 10.3390/insects13090777

**Published:** 2022-08-28

**Authors:** Xudan Kou, Shichao Bai, Yufeng Luo, Jiuyang Yu, Huan Guo, Chao Wang, Hong Zhang, Chunhuan Chen, Xinlun Liu, Wanquan Ji

**Affiliations:** 1State Key Laboratory of Crop Stress Biology for Arid Areas and College of Agronomy, Northwest A&F University, Yangling 712100, China; 2Shaanxi Research Station of Crop Gene Resources and Germplasm Enhancement, Ministry of Agriculture, Yangling 712100, China

**Keywords:** cage, life table, plant–aphid interaction, *Sitobion avenae*, small insect, *Triticum aestivum*

## Abstract

**Simple Summary:**

Clip cages are increasingly used in the study of plant–small insect interactions; however, the heavy weight and low light transmittance of clip cages cause damage to plants and affect insect growth and reproduction. This study aimed to develop a lightweight, transparent clip cage and investigate its effects on *Sitobion avenae* (Fabricius) and *Triticum aestivum*. The results indicate that the modified clip cages neither altered the *S. avenae* life-history parameters nor activated the *T. aestivum* defense responses and can, therefore, be used for plant–aphid interaction studies.

**Abstract:**

Clip cages are commonly used to confine aphids or other small insects to a single leaf when conducting plant–small insect interaction studies; however, clip cages are usually heavy or do not efficiently transmit light, which has an impact on leaf physiology, limiting their application. Here, simple, lightweight, and transparent modified clip cages were constructed using punched clear plastic cups, cut transparent polyvinyl chloride sheets, nylon organdy mesh, and bent duck-bill clips. These cages can be clipped directly onto dicot leaves or attached to monocot leaves with bamboo skewers and elastic bands. The weight, production time, and aphid escape rates of the modified clip cages were 3.895 ± 0.004 g, less than 3 min, and 2.154 ± 0.323%, respectively. The effects of the modified clip cage on the growth, development, and reproduction of the English grain aphid (*Sitobion avenae* Fabricius) in comparison with the whole cage were studied. The biochemical responses of wheat (*Triticum aestivum*) to the cages were also investigated. No significant differences were observed in the life table parameters, nymph mortality, and adult fecundity in *S. avenae* confined to clip cages and whole cages, but the clip cages were more time efficient than whole cages when conducting life table studies. Moreover, the hydrogen peroxide accumulation, callose deposition, and cell necrosis in wheat leaves covered by empty clip cages and empty whole cages were similar, and significantly lower than treatments where the aphids were inside the clip cage. The results demonstrate that the modified clip cages had negligible effects on the plant and aphid physiology, suggesting that they are effective for studying plant–small insect interactions.

## 1. Introduction

Aphids (Hemiptera: Aphididae) are phytophagous or polyphagous insects distributed worldwide [1,2,3]. Of the more than 5000 known species of aphids, about 450 species feed on cultivated plants, with hundreds of these causing a serious threat to grains, fruit trees, vegetables, and flowers all over the world [4,5]. As major pests, aphids can directly or indirectly reduce crop yield and quality by sucking on plants and transmitting about half of the approximately 700 insect-borne plant viruses [5,6]. *Sitobion avenae* (Fabricius), one of the most destructive and commonly occurring species in China, threatens the production of bread wheat (*Triticum aestivum*), one of the most important food crops in the world [2].

Life tables are the most effective and comprehensive tool for describing insect population dynamics [7]; however, traditional female, age-specific life tables ignore males in the population as well as stage differences, limiting their application [8]. Chi and Liu [9] and Chi [10] were the first to report the age-stage, two-sex life table theory. Most of the life table parameters, including net reproductive rate(*R*_0_), finite rate of increase (*λ*), intrinsic rate of increase (*r*), and mean generation time (*T*) were proposed in these two studies. Chi and Su [11] illustrated the concept and calculation method of age-stage life expectancy (*e_xj_*) when performing a life history study of *Aphidius gifuensis* and *Myzus persicae* in the laboratory. The calculation of the age-stage reproductive value (*v_xj_*) was reported for the first time in a study of *Spodoptera litura* populations under different conditions [12,13]. Presently, the age-stage, two-sex life table theory is considered the most detailed and accurate description of insect survival, development, fecundity, and stage differences [8]; therefore, the age-stage, two-sex life table has been widely used in diverse subject areas: for the effects of temperature, pesticide, or climate change on insect life history [14,15,16,17]; host plant antibiosis assays [18]; plant–pest–predator/parasitoid relations [19]; effects of microorganisms and endosymbionts [20]; inbreeding [21]; etc. [8].

Integrated pest management (IPM) techniques are important for pest control [22]. In IPM programs, it is necessary to understand the growth, development, and reproduction of pests under varied conditions of temperature, insecticide usage, and cultivars [23,24,25,26,27]. Evaluating the antibiosis of plants against aphids through no-choice tests, for example, life table studies, nymph mortality, and adult fecundity, is an important part of crop protection research [23,28,29,30]. In no-choice studies, aphids are generally confined to whole cages or clip cages, either individually or in groups [24,26,31,32,33,34]. Whole cages can cover all plants in a pot, while clip cages only accommodate a small portion of the leaves [35]. In whole cages, it is difficult to find a single aphid that is small and freely moves over the entire plant. Additionally, checking or removing molts and new nymphs from the entire plant is not easy [36]; therefore, clip cages are necessary in no-choice studies, especially for investigations of individual aphids. Clip cages are now widely used to study the effects of factors such as temperature, cultivar, carbon dioxide levels, soil nutrient status, and insecticide concentrations on the settlement, biomass, survival, and reproduction of small insects [26,34,37,38,39].

Clip cages and whole cages have been used in the studies of plant resistance to aphids over the past few decades [32,38,40,41,42]; however, some studies can only be performed using clip cages. For example, clip cages are ideal for systemic acquired resistance studies. Zhuang et al. [43] treated *Cuscuta australis* plants with 30 clip-cage-enclosed adult *M. persicae* for 24 h and collected the third trifoliate leaves of soybean and the enclosed tissues of *C. australis* for phytohormone quantification and transcriptomic analysis to study the systemic signaling between *C. australis* and soybean host under aphid infestation. In addition, clip cages have been used to study plant responses to sequential biotic stresses or the systemic effects of aphid pre-infestation on aphids, small insects, or pathogen attack [42,44]. Furthermore, clip cages facilitate the precise studies of plant responses to small insect infestations. For example, numerous studies have investigated aphid-infested leaves at the physiological, cellular, and molecular levels using clip cages [44,45,46,47]. To compare the effects of two aphid species, *Schizaphis graminum* and *S. avenae*, on wheat physiological and defense responses, aphids were transferred to wheat leaves and restricted in clip cages, and the leaves were collected after feeding to evaluate the hydrogen peroxide accumulation, total chlorophyll content, salicylic acid content, defense-related gene expression, and metabolic profiling [44]. Zhang et al. [47] placed clip cages, each containing 20 third instar *S. graminum*, on the second leaves of wheat plants and harvested the leaf tissues for hydrogen peroxide detection, chlorophyll levels detection, antioxidant enzymes determination, and transcriptome analysis to reveal the molecular mechanism of wheat resistance to *S. graminum.*

At present, clip cages can be divided into two principal types according to their basic components. One type of clip cage is mainly made of plastic tubes and metal hair clips, and the other is based on foam floating tubes [36,37,48,49,50]. The former is heavy, and many plants, especially monocots, cannot bear its weight. Additionally, it is difficult to make and may damage plant leaves. The main problem of the latter is poor light transmittance, which may interfere with the normal growth and development of both aphids and the plants. Furthermore, foam floating tubes allow aphids to escape more frequently, affecting the accuracy of the results. Moreover, no study has comprehensively explored the applicability of clip cages in aphid no-choice tests and aphid-infested plants studies.

The present study improved the classic plastic clip cage. This modified clip cage is economical, transparent, simple to make, and is suitable for monocot and dicot plants in various environments. Experiments were performed to verify whether the modified clip cage could be used in plant–aphid interaction research. These experiments were: (1) a comparison of the life table parameters of aphids confined by clip cages and whole cages (the aphids were studied individually); (2) a comparison of nymph survival and adult fecundity in clip cage and whole cage groups (the aphids were studied in groups); (3) a comparison of the cell necrosis, hydrogen peroxide accumulation, and callose deposition of wheat leaf segments covered with empty clip cages, empty whole cages, and clip cages with aphids. The advantages and limitations of the modified clip cage are also discussed.

## 2. Materials and Methods

### 2.1. Making the Clip Cage

STEP 1: Preparation of the materials and tools—clear plastic cups (bottom diameter: 2.8 cm, top diameter: 3.9 cm, and height: 3.4 cm; Weijillong Plastic Products Factory, Foshan, China); transparent polyvinyl chloride sheets (0.5 mm thickness; Taixinlong Plastic Products Co., Ltd., Dongguan, China); metal duck-bill clips (4.5 cm length; Jiye Trading Co., Ltd., Yiwu, China); nylon organdy mesh (80 holes per inch; Kangyijia Biotechnology Co., Ltd., Qingdao, China); elastic bands (M&G Chenguang Stationery Co., Ltd., Shanghai, China); bamboo skewers (Gold Miner Network Technology Co., Ltd., Hangzhou, China); soldering iron (Hengqinzhongting Information Technology Co., Ltd., Zhuhai, China); hot melt glue gun (Kraftwelle Industrial Co., Ltd., Hangzhou, China); hot melt glue stick (Soli Technology Co., Ltd., Beijing, China); non-stick pan (Fengwei Industrial Co., Ltd., Shanghai, China); scissors; ruler; pliers; art knife.

STEP 2: Pretreatment of the plastic cups, polyvinyl chloride sheets, and nylon organdy mesh—use a soldering iron to punch three holes in the bottom of the plastic cups. Cut a polyvinyl chloride sheet with an art knife into squares with sides of about 4.2 cm. Cut the nylon organdy mesh with a pair of scissors into squares with sides of about 3.5 cm (Figure 1A–C and Figure 2).

STEP 3: Pretreatment of the duck-bill clips—use pliers to bend the bottom arm of the duck-bill clips by 65–75 degrees along the joint, and buckle the other arm 10–20 degrees from the middle (Figure 1D and Figure 2).

STEP 4: Assembly and fixing—assemble the processed plastic cups, polyvinyl chloride sheets and duck-bill clips, and fine-tune the arm angle of the duck-bill clips. Use hot melt glue to secure the connection of the clip to the polyvinyl chloride sheet and plastic cup (Figure 1E–G and Figure 2).

STEP 5: Paste the nylon organdy mesh—lay the cut nylon organdy mesh on a non-stick pan. Coat the bottom of the cup with hot melt glue, and immediately place it on the prepared nylon organdy mesh and press tightly. After the hot melt glue has cooled and solidified, use the scissors to trim the excess nylon organdy mesh (Figure 1H–K and Figure 2).

### 2.2. Using the Clip Cage

STEP 1: Open the door of the clip cage and insert a finger tube or a simple suction-implement containing one or more aphids into the clip cage, and flick the wall of the tube to introduce the aphids.

STEP 2: Place the bamboo skewer at the corner between the hand-held part and the bottom arm of the duck-bill clip, and secure it with an elastic band (Figure 1L and Figure 2).

STEP 3: Attach the dorsal leaf surface to the polyvinyl chloride surface of the cage, and place the short end of the bamboo skewer close to the base of the leaf, then gently clamp the leaf with the clip cage, and then insert the bamboo skewer into the soil (Figure 3).

### 2.3. Plants and Aphids

A single colony of *S. avenae* was collected in 2019 from an unsprayed wheat field at the Northwest A&F University, Yangling, Shaanxi, China (108.07° E, 34.28° N), and was maintained on wheat plants (*T. aestivum* variety Aikang 58) in a climate chamber at 20 ± 1 °C, 50 ± 5% relative humidity, and a 16:8 day:night cycle for more than two years. Wheat seeds of Aikang 58 were soaked in double-distilled water for one day and germinated on moist filter paper to a height of about 2 cm in the dark at 25 ± 1 °C. Subsequently, healthy seedlings were selected and sown with three seedlings per pot (7 × 7 × 10 cm). The plants were cultivated in a growth chamber under the same conditions described above. All experiments were conducted at the two-leaf stage.

### 2.4. Life Table Study

All aphid bioassay experiments were divided into two groups: the whole cage group and the clip cage group. Except for the cages, both insect groups were treated and cultured identically. The clip cage was constructed as described in Section 2.1. The whole cage was constructed by cutting the bottom of a clear polyethylene terephthalate plastic bottle (diameter: 6.5 cm and height: 20 cm; Sutai Plastic Products Co., Ltd., Cangzhou, China) and gluing the nylon organdy mesh over the cut end. A single apterous adult was transferred to each cage, with one cage per plant, and 67 wheat plants in each group. All the adults were removed and only one first instar nymph per plant remained 24 h after infestation. From the next day, individual aphids were verified at 14:00 every day for nymph mortality and instar changes. Molts were removed until the nymphs developed into adults. Thereafter, the offspring of the adults were recorded and removed on a daily basis until the adults died, and the date of death was recorded.

### 2.5. Mortality and Weight of Nymphs

Ten wingless adults were introduced into each cage. After 24 h, the adults and excess nymphs were removed gently and 15 newborn nymphs were maintained per wheat plant. The number of aphids was checked and the mortality of nymphs was computed after 6 days. Furthermore, to investigate the weight of the aphids, the covered leaf segments along with the aphids and clip cages were cut and weighed. The aphids were brushed away, and the remainders were weighed again. An electronic balance (ME204E; Mettler Toledo Corp., Switzerland) with an accuracy of 0.1 mg was used for the weighing.

### 2.6. Adult Fecundity and Honeydew Content

Each wheat plant was infested with five 6-day-old aphids, and the aphid number was surveyed 6 days after infestation. Sixty-seven biological replicates per treatment were performed. To obtain the weight of the honeydew, the clip cages were numbered and weighed in advance. After the number of aphids was recorded, the debris inside the clip cages was removed, and the clip cages were weighed again with an electronic balance (ME204E).

### 2.7. Escape Rate

Clip cages containing newborn nymphs produced by 15 adults within 24 h were individually attached to the surface of the wheat leaves and covered by whole cages. After 5 days, the number of aphids in the clip cages and whole cages was counted, and the escape rate was calculated. Seventy plants were used in this experiment.

### 2.8. Histological Staining

For each experiment, the wheat plants were treated with empty clip cages, empty whole cages, and clip cages containing 30 second instar nymphs. Each treatment included at least 12 biological replicates. Samples for hydrogen peroxide accumulation, callose deposition, and cell necrosis assays were collected at 4 days, 4 days, and 14 days after treatments, respectively.

Trypan blue staining was performed based on the method described by Jones and Deverall [51] with some modifications. Briefly, leaf segments were placed into a boiling trypan blue staining solution [(0.01% trypan blue (*w/v*) in lactic acid:phenol:glycerol:water (1:1:1:1 *v/v*)):absolute ethanol = 1:1 (*v/v*)] for 8 min, hyalinized in saturated chloral hydrate for two days, and finally preserved in 50% glycerol. To visualize the hydrogen peroxide accumulation, the leaf segments were incubated according to the instructions of the Enhanced HRP-DAB Chromogenic Kit (Tiangen Biotech (Beijing), China) and clarified with decolorizing solution [ethanol:glacial acetic acid:glycerol (3:1:1 *v*/*v*)] for 30 min in a boiling water bath. The visualization was performed using an Olympus SZX16 microscope (Olympus Corporation, Tokyo, Japan). The hydrogen peroxide accumulation area and necrotic area were quantified using the ImageJ software version 1.53c (National Institutes of Health, Bethesda, MD, USA) with the thresholds of RenyiEntropy and Shanbhag, respectively [52].

The callose deposition was determined according to Bouwmeester et al. [53]. Briefly, the leaves were decolorized in absolute ethanol and stained overnight with aniline blue solution [0.01% aniline blue (*w/v*) in 150 mM phosphate buffer (pH = 8.0)]. Photographs were taken with an Olympus BX-43 microscope (Olympus Corporation). The fluorescence intensity and area were calculated with the ImageJ software using the following parameters: threshold = Yen; limit to threshold; mean gray value; show = overlay [52]. Ten photographs from each sample were randomly selected for processing. The relative callose content was 10,000 times the sum of the fluorescence area multiplied by the fluorescence intensity of each callose deposition site.

### 2.9. Statistical Analyses

The life table study of *S. avenae* was analyzed using the TWOSEX-MSChart program [54] based on the age-stage, two-sex life table theory [8,9,10]. The mean values and standard errors of adult longevity, adult pre-oviposition period (APOP), total pre-oviposition period (TPOP), oviposition days, fecundity, *R*_0_, *λ*, *r*, *T*, doubling time (*DT*), and gross reproduction rate (*GRR*) were estimated using the bootstrap method [55] with 100,000 replicates [56] based on the methods of [9,10]. The paired bootstrap test at a 5% significance level was performed to analyze the differences in the life table parameters. Moreover, other population parameters were calculated, including the age-stage survival rate (*s_xj_*), age-specific survival rate (*l_x_*), age-specific fecundity (*m_x_*), stable age-stage distribution (SASD) [9,10], *v_xj_* [12,13] and *e_xj_* [11]. The Kaplan–Meier method was used to draw survival curves, and the log-rank test was used to detect the difference in the survival rate between the two groups. *R*_0_ represents the average total number of offspring that an individual can produce in its life span. *λ* represents the growth rate of the population when time is infinite and the population reaches age-stage stable distribution. *r* represents the growth rate of the population in an unlimited environment. The population size increases at the rate of *e^r^* per time unit. *T* represents the time required for the population to grow to *R*_0_-fold of its size at the stable age distribution. *s_xj_* represents the possibility of a newborn nymph surviving to age *x* and stage *j*. *l_x_* shows the possibility of a newborn nymph surviving to age *x*. *m_x_* shows the mean fecundity of individuals at age *x*. *v_xj_* describes the contribution of individuals at age *x* and stage *j* to the future population. *e_xj_* demonstrates the expected survival time of individuals at age *x* and stage *j.*

The GraphPad Prism software version 8 (GraphPad Software, Inc., La Jolla, CA, USA) was used for the statistical analysis and graphing. In the nymph mortality and adult fecundity experiments, the statistical analysis was carried out using an unpaired *t*-test. In the histological staining, the differences among the groups were calculated by a one-way analysis of variance (ANOVA) followed by a Games–Howell’s multiple comparisons test. *p* < 0.05 was considered statistically significant. The data were expressed as mean ± SE.

## 3. Results

### 3.1. Effects of Clip Cages on the Antibiosis of Sitobion avenae

To elucidate the applicability of the clip cages in the no-choice study, a life table study was performed. The results (Table 1) showed that the biological parameters, including the developmental duration, adult longevity, total longevity, APOP, TPOP, oviposition days, and fecundity of *S. avenae* were not significantly different between the whole cage and clip cage groups. At the same time, significant differences were not detected between the two groups in the demographic parameters of *S. avenae*, such as *R*_0_, *λ*, *r*, *T*, *DT*, and *GRR* (Table 2). 

The overlapping developmental stages indicated the diversity in the developmental rates of aphid individuals in both the clip cage and whole cage groups (Figure 4A,B). The trends of the *s_xj_* curves were similar. Specifically, most aphids in these two cages had similar age-stage survival rates and longevity. All the newborn nymphs survived to the adult stage, and the survival rates of adult females decreased from the 32nd day in both groups. According to the *l_x_* curve, the survival rate of aphids in the clip cages was slightly higher than that in the whole cages from the point that the survival rate declined (Figure 4C). The *m_x_* and *l_x_m_x_* values of the whole cage group were higher than those of the clip cage group on days 6–8 and 12–21, while the trend was reversed on days 9–11 and 22–24, and the aphid fecundity and maternity were slightly lower in the whole cage group than in the clip cage group. Both groups displayed similar *m_x_* and *l_x_m_x_* after the 25th day (Figure 4F,I). At the same time, the maximal *m_x_* and *l_x_m_x_* values for both groups occurred on day 9. On day 22, small fecundity and maternity peaks occurred in both groups, but the peak height in the clip cage group was higher (Figure 4F,I). The survival analysis showed no significant difference in survival probability between the two groups (*p* = 0.075) (Figure 4L). The *v_xj_* (Figure 4D,E), *e_xj_* (Figure 4G,H), and SASD (Figure 4J,K) of the *S. avenae* in the clip cages followed a similar pattern to those in the whole cages. The highest unique age-stage reproductive peak for both cages appeared on day 9, and the maximum *v_xj_* values for the whole cage and clip cage groups were 14.809/day and 15.282/day, respectively (Figure 4D,E).

To further validate the applicability of the clip cages, the nymph mortality and adult fecundity were investigated. No significant difference was observed in the nymph mortality (*p* = 0.772) between the clip cage and whole cage groups, which were 8.615% ± 1.181% and 7.071% ± 0.984%, respectively (Figure 5A). For adult reproduction, five third or fourth instar nymphs produced 44.200 ± 0.987 and 45.985 ± 0.990 offspring within 6 days in the clip cage and whole cage groups, respectively (Figure 5B). Differences in adult fecundity between the two groups were not detected by the unpaired *t*-test (*p* = 0.202) (Figure 5B); thus, the clip cages can be used for aphid antibiosis research.

### 3.2. Clip Cages Facilitate Determination of Sitobion avenae Weight

Weighing aphids directly is not convenient or accurate due to their mobility; therefore, aphid weight can be obtained by subtracting the weight of the leaf segments and the clip cage after the removal of aphids from the total weight of the aphids, leaf segments, and the clip cage. In this study, the mean body weight of *S. avenae* on day 6 after birth was 0.507 ± 0.022 mg.

### 3.3. Clip Cages Facilitate Honeydew Collection and Quantification

Honeydew can be roughly quantified by the difference in cage weight before and after honeydew collection. In this study, five adult aphids secreted 16.009 ± 0.871 mg of honeydew over 6 days. In addition, honeydew can be washed for accurate quantification and compositional studies by spectrophotometry and chromatography, respectively.

### 3.4. Clip Cages Promote the Study of Plant Phenotypes after Aphid Infestation

To investigate the effect of the clip cages on the plant physiological phenotype, the following experiments were performed (Figure 6). In the trypan blue staining experiments, no significant differences in the cell necrotic areas were detected between the empty clip cage and empty whole cage groups, which were 0.016 ± 0.004 mm^2^ and 0.022 ± 0.003 mm^2^, respectively; however, the area of cell necrosis significantly increased after aphid infestation (3.311 ± 0.359 mm^2^). The leaf segments covered by the empty clip cages (0.091 ± 0.011 mm^2^) and the empty whole cages (0.117 ± 0.015 mm^2^) accumulated similar areas of hydrogen peroxide, which were significantly lower than those of the aphid-infested leaves (1.885 ± 0.271 mm^2^). Similarly, the relative callose content in the clip cage with aphids group (30.860 ± 1.062) was significantly higher than that in the empty clip cage group (7.244 ± 0.279) and the empty whole cage group (6.496 ± 0.479); thus, the clip cage can be used for the study of the plant phenotype after aphid infestation.

## 4. Discussion

This study reports the construction and application of modified clip cages made of plastic cups and metal duck-bill clips that can confine aphids or other small insects to certain parts of leaves. These cages facilitate estimating the antibiosis parameters of aphids, collecting honeydew, and observing the physiological phenotype of aphid-infested leaf segments. Compared with the classic clip cages [36], these modified clip cages allow for better attachment to monocots and are convenient to produce.

Plants defend themselves from insect attacks by a localized defense mechanism, namely, a hypersensitive response [57]. Most herbivorous attacks result in reactive oxygen species bursts, cell wall cross-linking, altered membrane permeability, callose deposition, and phytoalexin accumulation, ultimately leading to programmed cell death [57]. Histochemical staining was performed to investigate the effect of the clip cages on plant defense responses. The current work revealed that the hydrogen peroxide accumulation, callose deposition, and cell necrosis were similar between the clip cage and whole cage groups, but they significantly increased for the treatment groups of the clip cages with aphids (Figure 6). The results indicated that the clip cages did not cause oxidative bursts around the feeding sites and, thus, did not trigger changes in the plant physiology or secondary metabolites, whereas the aphids induced strong defensive responses. In addition, previous studies have shown that clip cages may alter leaf physiological characteristics, such as light saturation, CO_2_ exchange rate, soluble protein content, leaf area, leaf temperature, and chlorophyll content [58,59]; thus, the interpretation of the data might be compromised when the focus of a study is related to plant development. Further studies should be performed to confirm the changes in plant growth and development induced by the clip cages. 

The mechanisms of plant defense against aphids can be classified as antibiosis, which impedes aphid survival, growth, development, and fecundity; antixenosis, which influences aphid settlement and feeding behavior; tolerance, which is the ability of a plant to withstand or recover from the destruction of aphid populations [60]. Whole cages and clip cages are practical tools for aphid and other small insect antibiosis studies [23,28,41,61]; however, information on the effect of cages on small insects is scarce. In this study, a cellular hypersensitivity response was not activated in the leaf segments covered with the clip cages and the *S. avenae* life table parameters (Table 1 and Table 2), nymph mortality, and adult fecundity (Figure 5) were similar between the clip cage and whole cage groups. Consistent with these results, Mowry [62] found that the fecundity of two aphid species, the green peach aphid and corn leaf aphid, was similar between clip cages and whole cages. These observations suggest that resistance mechanisms may not be activated in clip-cage-covered leaves, which in turn do not affect aphid growth and reproduction, whether the aphids are studied individually or in groups; however, Taravati and Mannion [35] demonstrated that the adult longevity and fecundity of rugose spiraling whitefly were significantly reduced in whole cages compared with clip cages. The design of the clip cages (opaque, damage to leaves, etc.) or the bisexual reproduction of whiteflies may account for these differences. 

To elucidate the effects of the clip cages on aphid populations more comprehensively and accurately, an aphid age-stage specific life table curve was generated. Aphids feeding on plants covered by the clip cages exhibited similar *v_xj_*, *e_xj_*, and SASD compared with aphids feeding on whole-cage-covered plants (Figure 4). This indicates that the population structure, expected survival rate, and contribution to future aphid populations at various ages of the aphid population generation cycle were not affected by the cages; however, the *l_x_*, *m_x_*, and *l_x_m_x_* were slightly different between the two groups throughout the aphid life cycle (Figure 4). The sample size was not large enough that subtle differences in aphid reproduction and survival between the two groups would cause large differences in the life table curves, which might explain the slight difference. Moreover, the effects of various treatments on the fecundity and longevity of small insects have elsewhere been influenced by the cage setting [39,63,64]. For instance, neonicotinoid seed treatment resulted in a significant increase in spider mite density when whole cages were used, whereas similar populations were observed when soybean leaves were covered by clip cages [39]. Similarly, Sparks et al. [63] found a significant interaction between insecticides and bioassay methods in whitefly adult mortality, with a clip cage bioassay being the most sensitive to insecticide dose. 

The time spent every day in conducting the life table study was recorded, and it was found that the time saved by the clip cage group increased as the experiment progressed (Table 3). In addition, in the life table study, 67 replicates were set in both groups, but at the end of the experiment, only 58 valid replicates were left in the whole cage group, indicating that it is easier to manipulate aphids using clip cages rather than whole cages.

In the present study, we have improved the raw materials and production process of the classic plastic clip cage [36]. The materials required to make the modified clip cages were inexpensive and easily available. For instance, the price to make a modified clip cage was only CNY 0.29 (equivalent to GBP 0.03) in China, which was much lower than previous cages (material costs of GBP 0.16 and GBP 0.92 for the foam clip cage [49] and plastic clip cage [36], respectively). In addition, it took less than 3 min to make a modified clip cage, which was more than the time taken for the foam clip cage [49] (less than 2 min), but less than the time taken for the plastic clip cage [36] (up to 5 min). These advantages might be of great significance for the mass production of clip cages and the replacement of damaged cages. 

The escape rate of *S. avenae* from the modified clip cage was 2.154 ± 0.323% (Appendix A), which was significantly lower than the 6% of the foam clip cage [49] and the 40% of the plastic clip cage [36]; thus, more accurate results can be obtained using the modified clip cages than the previous clip cages. Furthermore, the average weight of the modified clip cage was only 3.895 ± 0.004 g (Appendix A), which can be accommodated by common dicots. For monocots (and some dicots), bamboo skewers can be used for supports, and the position of the clip cage on the bamboo skewers can be adjusted according to the position of leaves being studied. This means the cages can be used flexibly and widely adapted to a range of different plants. Clip cages are also easy to store and transport, as they are small, light, and strong, with detachable bamboo skewers, enabling their use in incubators, greenhouses, and fields. After an experiment, clip cages can be reused after removing the debris and cleaning their inner walls. Furthermore, modified clip cages can be adapted for other small insects, such as leafhoppers and whiteflies, by using nylon organdy cloth with a suitable mesh size.

## 5. Conclusions

An economical, lightweight, transparent, and widely applicable clip cage was designed and tested in this study. The life table parameters, nymph mortality, and adult fecundity were not significantly different between the clip cage and whole cage groups, suggesting that the clip cages can be used in colonization assays whether studying *S. avenae* individually or in groups. Importantly, the clip cages did not affect the plant physiological phenotypes, indicating that they do not activate wheat defense responses and can be used as a tool to study aphid-infested plants. In addition, the modified clip cages can be used to study a wide variety of plants and small insects and are suitable for different environmental conditions. This study provides an effective tool for the study of plant–small insect interactions.

## Figures and Tables

**Figure 1 insects-13-00777-f001:**
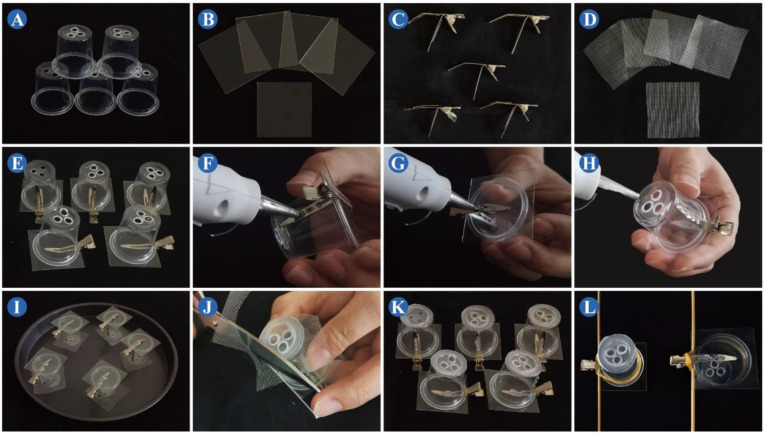
Process of making clip cages. (**A**) Plastic cups with holes in the bottom. (**B**) Cropped polyvinyl chloride sheets. (**C**) Bent duck-bill clips. (**D**) Cut nylon organdy mesh. (**E**) Assembly of materials in (**A**–**C**). (**F**) Apply hot melt glue to the junction of the clip and the plastic cup. (**G**) Glue the clip to the polyvinyl chloride sheet. (**H**) Apply a ring of hot melt glue to the bottom of the cup. (**I**) Seal the clip cage with nylon organdy mesh placed on a non-stick pan. (**J**) Trim the edges of nylon organdy mesh. (**K**) Finished clip cage. (**L**) Clip cage assembled with elastic bands and bamboo skewers.

**Figure 2 insects-13-00777-f002:**
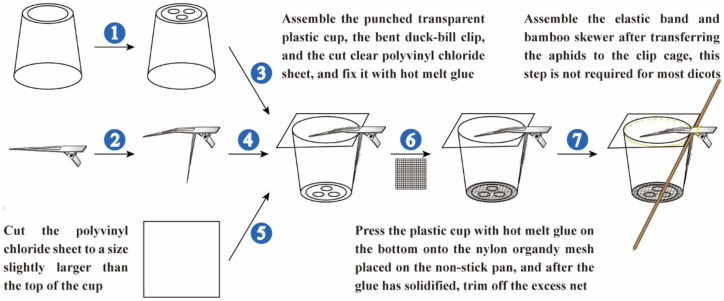
Demonstration of the steps to make a clip cage.

**Figure 3 insects-13-00777-f003:**
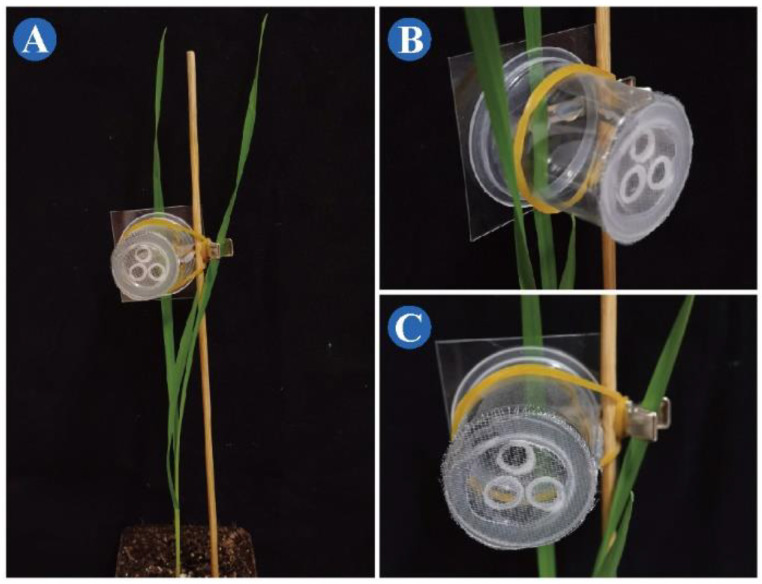
Modified clip cage in use. (**A**) Clip cage attached to *Triticum aestivum* leaves supported by bamboo skewers and elastic bands. (**B**) Side view of a clip cage in use. (**C**) Front view of a clip cage in use.

**Figure 4 insects-13-00777-f004:**
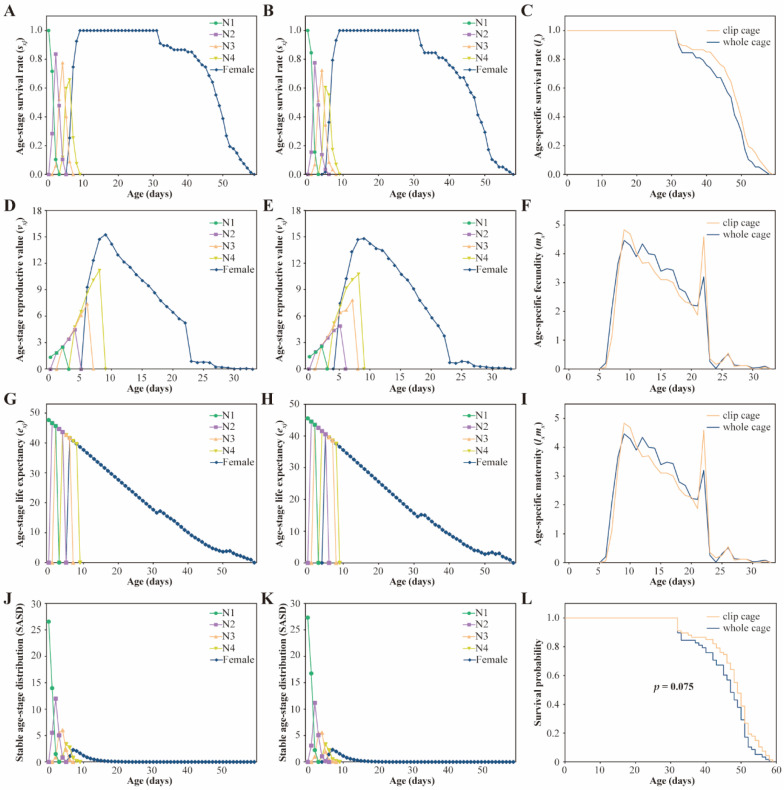
(**A**,**B**) Age-stage survival rate (*s_xj_*), (**C**) age-specific survival rate (*l_x_*), (**D**,**E**) age-stage reproductive value (*v_xj_*), (**F**) age-specific fecundity (*m_x_*), (**G**,**H**) age-stage life expectancy (*e_xj_*), (**I**) age-specific maternity (*l_x_m_x_*), (**J**,**K**) stable age-stage distribution (SASD), and (**L**) survival probability of *Sitobion avenae* confined to clip cages (**A**,**C**,**D**,**F**,**G**,**I**,**J**,**L**) and whole cages (**B**,**C**,**E**,**F**,**H**,**I**,**K**,**L**). Kaplan–Meier curves were constructed and log–rank (Mantel–Cox) tests were performed to detect differences in survival probability between the two cage groups. N1: first instar nymph; N2: second instar nymph; N3: third instar nymph; N4: fourth instar nymph.

**Figure 5 insects-13-00777-f005:**
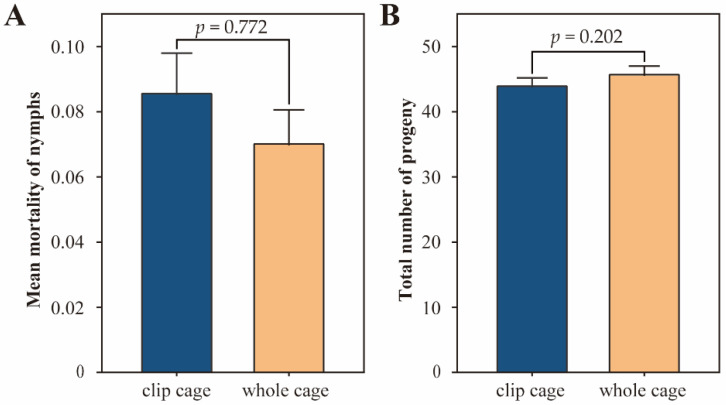
Clip cages did not significantly affect nymph mortality and adult fecundity of *Sitobion avenae*. (**A**) Mortality of 15 newborn nymphs reared for 6 days (*n* = 67). (**B**) Number of offspring produced by 5 adults during a 6 day reproduction period (*n* = 67). Differences were determined by an unpaired *t*-test.

**Figure 6 insects-13-00777-f006:**
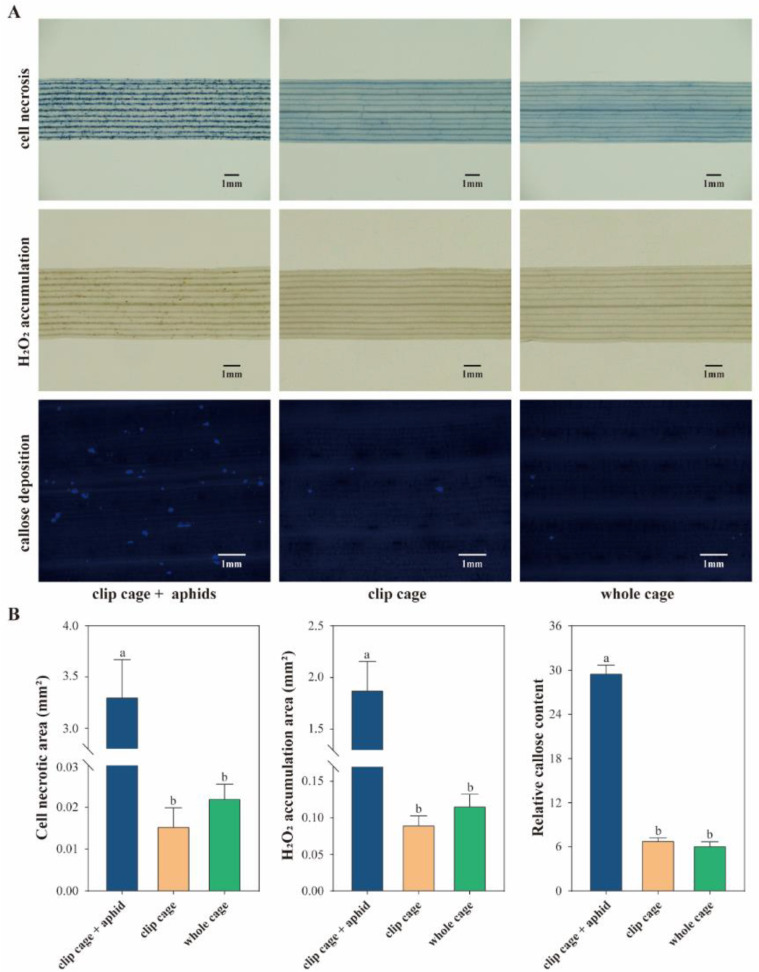
*Sitobion avenae*, but not the clip cages, significantly affected cell necrosis, hydrogen peroxide accumulation, and callose deposition. (**A**) Wheat leaves stained with trypan blue, 3,3′-diaminobenzidine, and aniline blue after being covered in different cages. (**B**) Cell necrotic area, hydrogen peroxide content, and relative callose content quantified using ImageJ software (*n* > 11). Different letters indicate significant differences at *p* < 0.05 level (one-way ANOVA, and Games–Howell’s multiple comparisons test).

**Table 1 insects-13-00777-t001:** Biological parameters of *Sitobion avenae* confined to clip cages and whole cages.

Biological Parameters	Whole Cage		Clip Cage		95% CI of Difference ^‡^	*p*-Value
	*n*	Mean ± SE ^†^	*n*	Mean ± SE ^†^		
First instar nymph (N1, days)	58	2.000 ± 0.073	67	1.821 ± 0.073	(−0.003, 0.362) ^ns^	0.053
Second instar nymph (N2, days)	58	1.586 ± 0.073	67	1.701 ± 0.067	(−0.042, 0.276) ^ns^	0.153
Third instar nymph (N3, days)	58	1.776 ± 0.060	67	1.851 ± 0.057	(−0.081, 0.231) ^ns^	0.348
Fourth instar nymph (N4, days)	58	1.534 ± 0.066	67	1.701 ± 0.060	(−0.010, 0.343) ^ns^	0.063
Pre-adult duration (days)	58	6.897 ± 0.119	67	7.075 ± 0.104	(−0.112, 0.476) ^ns^	0.239
Adult longevity (days)	58	38.655 ± 0.955	67	40.567 ± 0.868	(−0.813, 4.607) ^ns^	0.166
Total longevity (days)	58	45.552 ± 0.948	67	47.642 ± 0.864	(−0.606, 4.766) ^ns^	0.126
Adult pre-oviposition period (APOP, days)	58	0.793 ± 0.053	67	0.821 ± 0.047	(−0.117, 0.173) ^ns^	0.709
Total pre-oviposition period (TPOP, days)	58	7.690 ± 0.125	67	7.896 ± 0.099	(−0.099, 0.517) ^ns^	0.191
Oviposition days (days)	58	16.086 ± 0.206	67	15.806 ± 0.228	(−0.336, 0.896) ^ns^	0.374
Fecundity (offspring/female)	58	55.983 ± 1.442	67	53.284 ± 1.544	(−0.821, 6.174) ^ns^	0.132

^†^ Standard errors (SE) were estimated using the bootstrap procedure with 100,000 replications. ^‡^ Percentile confidence intervals of the differences between the clip cage and whole cage groups were calculated using the paired bootstrap test with 100,000 re-samplings. ^ns^ indicates no significant differences at *p* < 0.05 level.

**Table 2 insects-13-00777-t002:** Demographic parameters of *Sitobion avenae* confined to clip cages and whole cages.

Demographic Parameters	Whole Cage ^†^	Clip Cage ^†^	95% CI of Difference ^‡^	*p*-Value
Net reproductive rate (*R*_0_)	55.983 ± 2.641	53.284 ± 1.544	(−0.522, 10.567) ^ns^	0.089
Finite rate of increase (*λ*, days^−1^)	1.376 ± 0.009	1.362 ± 0.006	(−0.016, 0.025) ^ns^	0.687
Intrinsic rate of increase (*r*, day^−1^)	0.319 ± 0.007	0.309 ± 0.005	(−0.012, 0.018) ^ns^	0.686
Mean generation time (*T*, days)	12.607 ± 0.156	12.871 ± 0.130	(−0.192, 0.564) ^ns^	0.343
Doubling time (*DT*, days)	2.171 ± 0.051	2.244 ± 0.034	(−0.085, 0.139) ^ns^	0.679
Gross reproduction rate (*GRR*)	55.993 ± 1.442	53.290 ± 1.544	(−0.816, 6.179) ^ns^	0.131

^†^ Standard errors (SE) were estimated using the bootstrap procedure with 100,000 replications. ^‡^ Percentile confidence intervals of the differences between the clip cage and whole cage groups were calculated using the paired bootstrap test with 100,000 re-samplings. ^ns^ indicates no significant differences at *p* < 0.05 level.

**Table 3 insects-13-00777-t003:** Time spent in different stages of the *Sitobion avenae* life table study.

Stage ¶	Duration (Day)	Time Spent in Whole Cage Group ^†^ (Min)	Time Spent in Clip Cage Group ^†^ (Min)	Time Saved ^‡^ (Min)	*p*-Value ^§^
Entire	58	104.990 ± 8.302	26.467 ± 2.129	78.525 ± 9.071	<0.001 **
I	7	31.940 ± 5.284	11.493 ± 1.897	20.448 ± 3.967	0.007 **
II	21	67.784 ± 3.777	45.149 ± 2.466	22.635 ± 5.303	0.001 **
III	5	98.458 ± 5.668	29.747 ± 2.354	68.711 ± 6.256	0.002 **
IV	25	144.460 ± 11.296	16.981 ± 0.918	127.480 ± 10.560	<0.001 **

^†^ Time taken to study 50 aphids. ^‡^ Time saved = time spent in the whole cage group−time spent in the clip cage group. ^§^ Differences were assessed by paired *t*-test. ** indicates *p* < 0.01. ^¶^ Stages of aphid development. Entire, entire experimental period (entire = stage I + II + III + IV); I, nymph birth to 25% of aphids start reproducing; II, >25% of aphids start reproducing to 75% of aphids stop reproducing; III, >75% of aphids stop reproducing to all aphids stop reproducing; IV, all aphids stop reproducing to death of all aphids.

## Data Availability

The data presented in this study are available in the article or Appendix A.

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
