# Peer review of "Construction of a Modified Clip Cage and Its Effects on the Life-History Parameters of Sitobion avenae (Fabricius) and Defense Responses of Triticum aestivum"

_insects, 2022, doi:10.3390/insects13090777_

Round 1
Reviewer 1 Report
The scientific approach is sound but possibly could have been complemented by trialing another clip cage design along side your new design to see how it compares. Innovation like this is good to see and has a higher impact on future work that it probably gets credit for. Only had a few minor edits to include and some suggestions detailed below.
Minor Edits
Line 3 - half of “Sitobion avenae” not in italics in manuscript
Line 15 - remove “in turn” (redundant)
Line 22 - suggest: “leaf physiology; limiting their application.”
Line 26 – sentence starts with “And…” suggest rephrasing to not start with “and”. E.g . “The weight, production time, and aphid escape rates of the cages were…….., respectively”. Or emphasise that these are relatively low?
Line 27 – Remove “Moreover” (redundant)
Line 32 -33 – Unsure what this part of the sentence means, please make it clearer “
whereas the whole cages group took an extra 78.52 ± 9.071 minutes 32 per day in the life table study of 50 aphids”
Line 35: - suggest “… significantly lower than treatments where aphids were inside the clip cage.”
Line 48: what is meant by “dominant”? Most abundant, widespread? Please make it clearer. Remove “absolute”
Line 51 – remove comma in red “ To manage pests effectively, it is necessary…”
Line 55- 56 – “is one of the most important research areas” inside what frame of reference? Across all disciplines? Within aphid research? Plant health research? Crop protection? Make it more specific
Line 62-63 – Suggest: “In whole cages, it is difficult to find a single aphid that is small and able to move freely over the entire plant”
Line 65 – suggest replacing “irreplaceable” with “necessary”
Line 71 – Remove “Both” at the start of the line (redundant)
Line 74 – remove “the…..”
Line 95 – change to “Furthermore, foam floating tubes allow aphids to escape more frequently, affecting the accuracy of the results”
Line 114 - A google search suggests that “duck-bill clip” would be a better term than “duck tongue clip” suggest reviewing and replacing “duck tongue” throughout the document
Line 116 – Looks line any non-stick pan will work?
Line 138 – Leather bands = elastic bands I think. Please replace with elastic band throughout the document if correct. (also in fig. 2.)
Fig 2 text – replace “trashy net” with “excess net”
Line 146-148 – This step probably needs a better diagram to demonstrate the attachment of the elastic band. Figure 1L is not really showing much so could be replaced with a close up of the elastic band attachment
Line 173 – replace “bases” with “basis”
Line 227 – replace “significant level” With “significance level”
Line 255 – Avoid starting a sentence with “And…”. Remove “And…” and start with “All the newborn…..”
Line 258 – replace “since the survival rate declined” with “from the point the survival rate declined”
Line 303 – replace “leave segments” with “leaf segments”
Line 380 – Looks like part of the sentence is in a smaller font? “….with the clip cage….”
Line 396- 400 – sentence structure needs to be change to parse e.g. “………the clip cage and whole cage groups; but significantly increased for treatments groups of clip cages with aphids in…”
Line 413 – needs a capital at the beginning of the sentence “The average price….”
Line 428 – sentence needs revising “ For monocots (and some dicots), clip cages can be adapted by using bamboo sticks as supports……”
Line 430 – sentence structure suggest editing to : “ … leaves that are being studied. This means the cages can be used flexibly and widely adapted to a range of different plants.”
Line 434 – don’t start with “And” just remove it e.g. “After the experiment is complete……”
Line 436 – “leafhoppers”
Line 440 – suggest changing to “… clip cage was designed and tested.”
Author Response
Dear Reviewer,
On behalf of my co-authors, we thank you for giving us an opportunity to revise our manuscript, we greatly appreciate your positive and constructive comments and suggestions on our manuscript entitled “Construction of a modified clip cage and its effects on life-history parameters of Sitobion avenae (Fabricius) and defense responses of Triticum aestivum” (Manuscript ID: insects-1853536). We have studied the comments carefully and have made revisions which are marked in red on the paper. We have tried our best to revise our manuscript according to the comments. Attached please find the revised version, which we would like to submit for your kind consideration. We appreciate your warm work earnestly and hope that the correction will meet with approval. Looking forward to hearing from you.
With best wishes,
Yours sincerely,
Wanquan Ji and Xinlun Liu.

Reviewer 2 Report
The manuscript entitled “Production of a modified clip cage and its effects on life-history parameters of Sitobion avenae (Fabricius) and defense responses of Triticum aestivum” (Insects-1853536) brought results that need revision. The study investigated the potential use of clip cages as an effective tool for studying plant-small insect interactions, their impacts on the growth, development, and reproduction of the aphid (Sitobion avenae Fabricius) and the biochemical responses of wheat (Triticum aestivum). However, the manuscript lacks some important details for the reader/reviewer to fully understand. In addition, it is not well highlighted the importance of the life table parameters study and how the finds could be helpful for the management of Sitobion avenae. There are many confusing sentences throughout the manuscript, which are hard to understand. Please take into account that many sentences need to be rephrased, in my opinion. For these reasons, I strongly suggest to revise and improve this manuscript. I have listed some comments in the file attached.
Line 26-32: “And its weight, production time, and aphid escape rate were 3.895 ± 0.0038 g, less than 3 minutes, and 2.154 ± 0.323%, respectively”. Please check the datasets' digits as the datasets' presentations are in different digits both in text and tables. Consistency in the digits presentation of datasets is required.
Further in which table datasets are presented. Please present the data table without mentioning it in the abstract portion and also give the table number in text …..line 422- 427 or as a supplementary table if necessary.
Line 81: “aphid infestation on the secondary attack by aphids or small insects or pathogens”. Please rephrase the sentence
Line 82-84: The sentence “Furthermore, clip cages are now widely used in histological studies, omics studies or to study molecular responses of plants to small insects, such as gene expression and metabolite accumulation” is not clear to the readers. Please rephrase the sentence with relevant references. The references are too many.
Line 100-101: “cheap, transparent, simple and convenient to make, easy and flexible to use, and is suitable for monocotyledonous and dicotyledonous plants and various environments”. This sentence is too lengthy. Please rephrase the sentence.
Introduction regarding the life table study parameters is not included in the introduction portion. Please add some introduction regarding the life table parameters study.
Line 113: “transparent PVC plastic sheet”. The abbreviations used are not clear to the readers
Line 156-158: “S. avenae used in this study was collected from a wheat field” Either the field of the aphid collection site was sprayed or unsprayed.
Line 156-162: The year of the aphid collection is not mentioned in the text
Line 158: (T. aestivum var. Bainongaikang58). Please check the variety name if this is written correctly
Line 179-180, 185: In the sentence “to investigate the weight of aphids” which instrument was used to weigh the aphids and weight of honeydew. Please also add the company name and reference.
Line 201: Please explain the abbreviations used in the text for example w/v, and v/v)
Line 222: oviposition days (Od), Please check the abbreviation if this is correct
Line 240: Data were expressed as Mean ± SEM. Explain SEM
Line 282-283: For adult reproduction, five third or fourth instar nymphs reproduced 44.20 ± 0.9874 and 45.99 ± 0.9897” Please make the revisions as there are some datasets in two digits and others are in three. Please also mention where you presented these datasets (figures or tables). There may be a uniform format to present the datasets. This may be revised in the whole manuscript or where necessary
Line 320-322: Similar results were achieved………empty whole cage group (6.496 ± 0.4787). Some data is presented in two digits and others in three digits. Please revise this in the same format. This sentence is lengthy and confusing. Rewrite the sentence in the correct format.
Table S1: 127.5 ± 10.56. Please make the revisions as some datasets are in two digits and others are in three.
Table S1 may include in the manuscript as table 3 without mentioning the supplementary table. These changes may also be revised in the manuscript if necessary
Mention the details of the stages (1-IV) in the portion of the legend as this is not clear to the readers
Figure 2. Please replace the legends as Demonstration of the steps to make a clip cage
Figure 4. The authors presented the 4 nymphal instars of the male populations of aphids and females. Is the datasets also include the 4 nymphal instars of the female populations. Please explain the presentation of the datasets and why the authors don’t present the female nymphal instars.
Figure 4, 5 and 6: The legends of the figures are lengthy please elaborate on the figure legends in the precise text
Figure 6: Please correct the legends of the figures “The area of the H2O2 accumulation“ Delete the word “The” from the legends. Please correct the whole figure or where necessary.
Line 262: (Fig. 4F, I) please write in the correct format as (Fig. 4F, 1). This correction may be revised in the whole manuscript
Line 341-342: The sentence is confusing, please rewrite the sentence precisely. “and feeding behavior; and tolerance”
Line 343-345: References are too many. Please write precise and relevant references
Line 353-354: The sentence is not clear. Please rephrase the sentence precisely. These observations suggest that resistance mechanisms ….. were studied individually or in groups
Line 380-381: “with the clip cage 380 bioassay being the most sensitive to insecticides dose”. Formatting of the text is not the same and may be according to the journal’s style formatting and authors instructions
Line 382-384: Table S1 may include in the manuscript as table 3 without mentioning the supplementary table. These changes may also be revised in the manuscript
Line 388: (A. Lal et al., 2018). Please rewrite the reference in the journal’s style format. This may also be revised in the whole manuscript
Line 404: delete the word (CER) in the sentence to maintain uniformity in the sentence.
Line 402-405: The sentence is too lengthy, “Additionally, previous studies…..while leaf temperature and chlorophyll content increased”. Please rewrite the sentence in simple and more precise form.
Line 411-415, 416-420: The sentences are too lengthy and confusing. “The materials required for making the modified clip cages … per Haas et al. (2018)”. “the average price of…...
Please rewrite the sentence precisely in simple words. The reference may also be rearranged according to the journal’s style format.
Line 422-424: Rewrite the sentences in more clear writing. “The escape rate of S. avenae from….designed plastic clip cage
Line 427-431: The sentence is too lengthy” For monocots, clip cages are available……applicable to various plant leaves. Please rewrite the sentence in precise form.
Line 432: delete the word “and”
Line 431-434: The sentence is too lengthy, “In addition, clip cages are easy to store and transport….as incubators, green houses, and fields. Rewrite the sentence in accurate and precise form.
Line 434-435: “And after the experiment is completed, the clip cages can be reused after the debris on their inner walls is cleaned”. Please rephrase the sentence.
References style used in the manuscript can be listed according to the journal style with consistency. Follow the journal style formatting for reference arrangements throughout all references. Please also correct the formatting style according to the author’s instructions of the journal. Please check for typos and missing italics, double spaces, abbreviations etc.
Author Response

(The authors gave the same response as above.)

Reviewer 3 Report
In the work “Production of a modified clip cage and its effects on life-history parameters of Sitobion avenae (Fabricius) and defense responses of Triticum aestivum”, Kou et al. describe a new clip-cage to confine aphids and other small animals on plant leaves. They compare aphid and plant parameters using their clip-cages or a whole cage setup (where the aphids are free to move on the plant) to confine aphids. In resume, they detect no negative effect of the clip-cages on aphids or plant. All in all, the work is technically sound and the results show clearly that there is no effect of the clip-cages on aphids or plants.
However, I feel that the manuscript contains some points that should be addressed and that are listed hereafter:
I had some difficulties understanding Figure 4: Please explain better the different parameters. For example, I do not understand what is “age-specific maternity” in Fig 4I. Further, what is the difference between Fig 4C and Fig 4L (the plots are almost identical), and between Fig 4F and Fig 4I (the plots are identical)? Yet you use different terms in the y-axis. You should also please define the parameters separately in the text in the paragraph starting l 252.
Concerning Fig 6, you should use the same scale for all images. Concerning quantification by ImageJ, you should describe in detail how the quantifications were made. What you write in the Materials & Methods section, is not sufficient to understand quantification procedures. A better term for "death area" might be "necrotic area".
Indicating the sources of the material to assemble the clip-cages is recommended for technical papers like this.
l 113 Please verify if it is 5 mm thickness
Please describe more in detail the whole cages (size, material, provider).
I never heard of “life table assay” Could there be a more appropriate term?
l l68 How many aphids did you place per cage?
Paragraph starting l 301. This sentence starts without a context. Please reformulate the sentence or add a title as for honeydew weighing.
l 348 ...“did not activate plant defenses”. Because you did not measure hormones, you should precise “as judged by cell mortality, H2O2 production and callose deposition”.
l 414 please add for which geographic region the prices are valid.
Please describe what kind of balance you used, as this is critical for some experiments.
Since your main objective is to show aptitude of the new clip-cages for aphid and plant studies, the discussion can be shortened considerably without affecting quality of the paper.
Author Response

(The authors gave the same response as above.)

Round 2
Reviewer 2 Report
The manuscript entitled “Construction of a modified clip cage and its effects on life-history parameters of Sitobion avenae (Fabricius) and defense responses of Triticum aestivum” (Insects-1853536) carried results that need some corrections/revision. I have listed some comments.
Line 26-27: 2.154 ± 0.323%,” values are missing or not visible in the table, what is 0.323%., please write clearly in the supplementary table, if necessary
Line 126-132: ASC99332, L5-A10040; QT2654; SL-RJ730; ZY6205; Please delete the codes as already enough information, and some materials are without codes
Line 184: replace the 2.1. with 2.1. (without italics)
Line 350: Replace the word “whole” with a suitable word
Line 436-438: For monocots (and some dicots), ….leaves that are being studied. The sentence is lengthy please write precisely.
Line 456-457: Table S1. Details of S. avenae escape rate investigation. Table S2. Weight of modified clip cages. The supplementary table is missing or not visible in the datasets.
Please check for typos and missing italics, double spaces, abbreviations, etc.
Author Response
Dear Reviewer,
On behalf of my co-authors, we thank you for giving us an opportunity to revise our manuscript, we greatly appreciate your positive and constructive comments and suggestions on our manuscript entitled “Construction of a modified clip cage and its effects on life-history parameters of Sitobion avenae (Fabricius) and defense responses of Triticum aestivum” (Manuscript ID: insects-1853536). We have studied the comments carefully and have made revisions which are marked in red on the paper. We have tried our best to revise our manuscript according to the comments. Attached please find the revised version, which we would like to submit for your kind consideration. We appreciate your warm work earnestly and hope that the correction will meet with approval. Looking forward to hearing from you.
With best wishes,
Yours sincerely,
Wanquan Ji and Xinlun Liu
Response to Reviewer 2 Comments
Dear Editors and Reviewers,
Thank you for your replies and comments concerning our manuscript entitled “Construction of a modified clip cage and its effects on life-history parameters of Sitobion avenae (Fabricius) and defense responses of Triticum aestivum” (Manuscript ID: insects-1853536). These comments were all valuable and very helpful for revising and improving our paper, as well as providing important guidance for our future research. We have studied the comments carefully and made appropriate corrections, which we hope meet with your approval. The revised sections can be seen in the manuscript with track changes. The main corrections in the paper and the responses to the reviewers’ comments are as follows:
Point 1: Line 26-27: 2.154 ± 0.323%,” values are missing or not visible in the table, what is 0.323%., please write clearly in the supplementary table, if necessary
Response 1: We accept this comment. We have revised it.
Table S1 and Table S2 have been supplemented to Line 455-456. Table S1, Table S2, and the manuscript main document have been added to a zip file named insects-1853536-revised_round2 and uploaded to the MDPI system. Table S1 is named “Table S1. Details of Sitobion avenae escape rate investigation.xlsx”; Table S2 is named “Table S2. Weight of modified clip cages.xlsx”.
Point 2: Line 126-132: ASC99332, L5-A10040; QT2654; SL-RJ730; ZY6205; Please delete the codes as already enough information, and some materials are without codes
Response 2: We accept this comment. We have revised it.
Line 127-132 – “ASC99332, L5-A10040; QT2654; SL-RJ730; ZY6205;” has been deleted from the original manuscript. (Line 127-131)
Point 3: Line 184: replace the 2.1. with 2.1. (without italics)
Response 3: We accept this comment. We have revised it.
Line 184 - “2.1” has been replaced by “2.1” (Line 183)
Point 4: Line 350: Replace the word “whole” with a suitable word
Response 4: We accept this comment. We have revised it.
Table 3. - “Whole” has been replaced by “Entire”
Line 350 - “Whole, whole experimental period” has been replaced by “Entire, entire experimental period (entire = stage I + II + III + IV)” (Line 349-350)
These changes were based on the description by Yao et al. (2022) (Further, arsenic level in AsHepG2 cells reaches a plateau after six hours of treatment, whereas arsenic continues to increase in HepG2 cells during the entire experimental period.) and Schubert et al. (2022) (Total feed intake, and intake of MR, body weight (BW), and average daily gain (ADG) were not significantly different between MR22 and MR19 during the entire experimental period.).
References
- Yao, Dingyan et al. “AQP9 (Aquaporin 9) Determines Arsenic Uptake and Tolerance in Human Hepatocellular Carcinoma Cells In Vitro.” Cureus vol. 14,7 e26753. 11 Jul. 2022, doi:10.7759/cureus.26753
- Schubert, Dana Carina et al. “Impacts of Reducing Protein Content in Milk Replacer on Growth Performance and Health of Young Calves.” Animals: an open access journal from MDPI vol. 12,14 1756. 8 Jul. 2022, doi:10.3390/ani12141756
Point 5: Line 436-438: For monocots (and some dicots), ….leaves that are being studied. The sentence is lengthy please write precisely.
Response 5: We accept this comment. We have revised it.
Line 436-438 - “For monocots (and some dicots), clip cages can be adapted by using bamboo sticks as supports and the position of the clip cage on the bamboo skewers can be adjusted according to the plant height and the position of leaves that are being studied.” has been replaced by “For monocots (and some dicots), bamboo skewers can be used for supports, and the position of the clip cage on the bamboo skewers can be adjusted according to the position of leaves being studied.” (Line 435-437)
Point 6: Line 456-457: Table S1. Details of S. avenae escape rate investigation. Table S2. Weight of modified clip cages. The supplementary table is missing or not visible in the datasets.
Response 6: We accept this comment. We have revised it.
Table S1, Table S2, and the manuscript main document have been added to a zip file named insects-1853536-revised_round2 and uploaded to the MDPI system. Table S1 is named “Table S1. Details of Sitobion avenae escape rate investigation.xlsx”; Table S2 is named “Table S2. Weight of modified clip cages.xlsx”.
Point 7: Please check for typos and missing italics, double spaces, abbreviations, etc.
Response 7: We accept this comment. We have revised it.
Line 11, 461 – Double spaces have been deleted from the original manuscript.
Line 127 – “M&G Chenguang Stationery Co.,Ltd.” has been replaced by “M&G Chenguang Stationery Co., Ltd.” (Line 127) [Space has been added].
With best wishes,
Yours sincerely,
Wanquan Ji and Xinlun Liu.
Reviewer 3 Report
Some suggestins to improve the language:
Line 42 100 → hundreds
Line 64 and so on → and many more
Line 217 4 days, 8 days and 14 days?
Line 376 sucking site → feeding sites
Line 425 Is this British pound?
Line 442 reused on → reused after
Author Response
Dear Reviewer,
On behalf of my co-authors, we thank you for giving us an opportunity to revise our manuscript, we greatly appreciate your positive and constructive comments and suggestions on our manuscript entitled “Construction of a modified clip cage and its effects on life-history parameters of Sitobion avenae (Fabricius) and defense responses of Triticum aestivum” (Manuscript ID: insects-1853536). We have studied the comments carefully and have made revisions which are marked in red on the paper. We have tried our best to revise our manuscript according to the comments. Attached please find the revised version, which we would like to submit for your kind consideration. We appreciate your warm work earnestly and hope that the correction will meet with approval. Looking forward to hearing from you.
With best wishes,
Yours sincerely,
Wanquan Ji and Xinlun Liu
Response to Reviewer 3 Comments
Dear Editors and Reviewers,
Thank you for your replies and comments concerning our manuscript entitled “Construction of a modified clip cage and its effects on life-history parameters of Sitobion avenae (Fabricius) and defense responses of Triticum aestivum” (Manuscript ID: insects-1853536). These comments were all valuable and very helpful for revising and improving our paper, as well as providing important guidance for our future research. We have studied the comments carefully and made appropriate corrections, which we hope meet with your approval. The revised sections can be seen in the manuscript with track changes. The main corrections in the paper and the responses to the reviewers’ comments are as follows:
Point 1: Line 42 100 → hundreds
Response 1: We accept this comment. We have revised it.
Line 42 – “100” has been replaced by “hundreds” (Line 42)
Point 2: Line 64 and so on → and many more
Response 2: We accept this comment. We have revised it.
Line 64 – “and so on” has been replaced by “and many more” (Line 64)
Point 3: Line 217 4 days, 8 days, and 14 days?
Response 3: We accept this comment. We have revised it.
Line 217 – “4 days, 4 days, and 2 weeks” has been replaced by “4 days, 4 days, and 14 days” (Line 216)
Point 4: Line 376 sucking site → feeding sites
Response 4: We accept this comment. We have revised it.
Line 376 – “sucking sites” has been replaced by “feeding sites” (Line 375)
Point 5: Line 425 Is this British pound?
Response 5: We are very grateful for your valuable questions. Here's our explanation:
The British pound symbol is £. Currency representation in this manuscript was based on Haas et al. (2018). The reason we converted CNY into British pounds is that the price of the previous cages was calculated in British pounds.
Moreover, Line 424-426 – “For instance, the price to make a modified clip cage was only £0.03 in China, which was much lower than previous cages (material costs of £0.16 and £0.92 for foam clip cage [49] and plastic clip cage [36], respectively).” has been replaced by “For instance, the price to make a modified clip cage was only ¥0.29 (equivalent to £0.03) in China, which was much lower than previous cages (material costs of £0.16 and £0.92 for foam clip cage [49] and plastic clip cage [36], respectively).” (Line 423-426)
References
Haas, J.; Lozano, E.R.; Poppy, G.M. A simple, light clip-cage for experiments with aphids. Agric. For. Entomol. 2018, 20, 589-592.
Point 6: Line 442 reused on → reused after
Response 6: We accept this comment. We have revised it.
Line 442 – “reused on” has been replaced by “reused after” (Line 440)
With best wishes,
Yours sincerely,
Wanquan Ji and Xinlun Liu.